# Optimized In Vitro CRISPR/Cas9 Gene Editing Tool in the West Nile Virus Mosquito Vector, *Culex quinquefasciatus*

**DOI:** 10.3390/insects13090856

**Published:** 2022-09-19

**Authors:** Tran Zen B. Torres, Brian C. Prince, Alexis Robison, Claudia Rückert

**Affiliations:** Department of Biochemistry and Molecular Biology, College of Agriculture, Biotechnology & Natural Resources, University of Nevada, Reno, NV 89557, USA

**Keywords:** mosquitoes, CRISPR/Cas9, gene editing, *Culex*

## Abstract

**Simple Summary:**

*Culex* mosquitoes are responsible for many established and emerging mosquito-borne diseases worldwide. These mosquitoes serve as the principal global transmission vector of West Nile virus, which is the leading cause of mosquito-borne disease in humans in the United States. Unfortunately, effective therapeutic drugs and vaccines are still lacking for most mosquito-borne diseases, and current vector controls (e.g., insecticide use) have fallen short of eradicating the disease burden. These concerns highlight the importance of developing novel strategies to prevent arbovirus transmission by *Culex* mosquitoes, especially as the prevalence of *Culex-*borne diseases increases globally. Underlying such approaches is often an improved understanding of virus–mosquito interactions. To investigate the putative antiviral role of *Culex* immune genes, we developed and characterized a *Culex*-optimized CRISPR/Cas9 plasmid for use in *Culex quinquefasciatus*-derived (Hsu) cell cultures. We showed that this newly constructed plasmid allows for efficient and reliable codelivery of all CRISPR reagents in vitro in a single plasmid system. These findings suggest that this tool may serve as a valuable resource for the establishment of mutant Hsu-derived cell populations, enabling the identification of mosquito host genes involved in antiviral response. Elucidating gene functions supports the development of alternative gene-based vector control strategies for *Culex* mosquitoes.

**Abstract:**

*Culex quinquefasciatus* mosquitoes are a globally widespread vector of multiple human and animal pathogens, including West Nile virus, Saint Louis encephalitis virus, and lymphatic filariasis. Since the introduction of West Nile virus to the United States in 1999, a cumulative 52,532 cases have been reported to the CDC, including 25,849 (49.2%) neuroinvasive cases and 2456 (5%) deaths. Viral infections elicit immune responses in their mosquito vectors, including the RNA interference (RNAi) pathway considered to be the cornerstone antiviral response in insects. To investigate mosquito host genes involved in pathogen interactions, CRISPR/Cas9-mediated gene-editing can be used for functional studies of mosquito-derived cell lines. Yet, the tools available for the study of *Cx. quinquefasciatus*-derived (Hsu) cell lines remain largely underdeveloped compared to other mosquito species. In this study, we constructed and characterized a *Culex*-optimized CRISPR/Cas9 plasmid for use in Hsu cell cultures. By comparing it to the original *Drosophila melanogaster* CRISPR/Cas9 plasmid, we showed that the *Culex*-optimized plasmid demonstrated highly efficient editing of the genomic loci of the RNAi proteins Dicer-2 and PIWI4 in Hsu cells. These new tools support our ability to investigate gene targets involved in mosquito antiviral response, and thus the future development of gene-based vector control strategies.

## 1. Introduction

Arthropod-borne viruses (arboviruses) are responsible for many established and emerging infectious diseases globally, of which mosquito-borne diseases make up a significant portion [1]. The dramatic emergence of epidemic arbovirus diseases has been attributed to human activities. For instance, increased global air travel and the establishment of seaborne trade routes allowed for the wide global dispersion of *Aedes aegypti*, the major urban vector of dengue virus (DENV) [2]. DENV is now considered to be the most rapidly spreading mosquito-borne virus worldwide, with frequent outbreaks occurring in Southeast Asia, the Western Pacific Islands, Africa, and Latin America [3]. In the continental U.S., West Nile virus (WNV) is considered the most epidemiologically significant arbovirus [4] and is endemic and emerging in large parts of continental Europe [5]. Since its introduction to New York City in 1999 [6], WNV has become endemic throughout much of the U.S., and 52,532 human infections have been reported in all 48 contiguous states and Washington D.C. [CDC]. The southern house mosquito, *Culex quinquefasciatus*, has been identified as the primary urban vector of WNV in the southwestern U.S. [7], along with *Culex pipiens* and *Culex tarsali*s [8]. *Culex* spp. mosquitoes are also responsible for the emergence of several mosquito-borne viruses worldwide, including Usutu virus (USUV) and WNV in Europe [9], Japanese encephalitis virus in Asia [10,11], and Saint Louis encephalitis virus in the Americas [12,13]. Despite the growing public health threat that mosquito-borne viruses present, definitive treatments and/or safe and effective vaccines are lacking for most of these pathogens [14]. Although the use of chemical insecticides has significantly reduced the burden of mosquito-borne diseases in the twenty-first century [15], rising insecticide resistance of mosquitoes challenges this vector control strategy [16,17]. These issues highlight the importance of generating novel approaches to block the transmission of arboviruses by mosquitoes.

Arboviruses generally undergo transmission cycles between vertebrate hosts and arthropod vectors. To complete their transmission cycles, mosquito-borne viruses must overcome the mosquito antiviral immune responses. Despite the activation of such antiviral responses, arboviruses tend to establish persistent infection without inducing significant pathology in their mosquito vectors [18]. This suggests that infected mosquitoes possess efficient pathways to tolerate virus infection despite lacking the mammalian adaptive immune system. RNA interference (RNAi) plays a major role in controlling arbovirus infection through the degradation of viral RNA, and this topic has been reviewed comprehensively elsewhere [19]. Briefly, RNAi can be further divided into three silencing pathways: the microRNA (miRNA), the small interfering RNA (siRNA), and the P-element-induced Wimpy testis (PIWI)-interacting (piRNA) pathways. The miRNA pathway plays significant roles in post-transcriptional regulation of gene expression, but is not known to directly target arbovirus replication. In contrast, the siRNA pathway has long been considered the cornerstone of antiviral immune response in mosquitoes [20]. This pathway is initiated when mosquito cells detect exogenous virus-derived dsRNA, which is then cleaved and degraded into siRNAs by an endonuclease called Dicer-2 (Dcr-2). The siRNAs are subsequently loaded into the RNA-induced silencing complex (RISC) to degrade complementary viral RNA. Finally, the piRNA pathway primarily safeguards the germline from genomic disruption through regulation of transposon activity, which is highly conserved in most organisms [21]. In the model insect *Drosophila melanogaster*, the biogenesis of piRNAs begins with the primary processing pathway, where piRNA precursors are transcribed from piRNA clusters and processed into 24–30 nucleotide primary piRNAs [22]. These primary piRNAs then undergo secondary amplification through the binding and processing of transposon RNA. Unlike the *D. melanogaster* piRNA pathway [23], *Aedes* spp. and *Culex* spp. mosquitoes have also been shown to produce virus-derived piRNAs (vpiRNAs) in response to the replication of a variety of arboviruses (reviewed in [24]). Moreover, *Aedes* PIWI4 appears to have a role in mediating antiviral functions [25,26,27]. These additional functions of the mosquito piRNA pathway may be attributed to the expansion and somatic expression of multiple *piwi* genes [28,29,30,31]. However, the antiviral role of the piRNA pathway in mosquitoes remains poorly understood. Elucidating the potential antiviral properties of PIWI proteins will help in understanding the vector immune response of different mosquito species.

The discovery of clustered regularly interspaced short palindromic repeats (CRISPR)-associated protein 9 (Cas9), a powerful gene editing tool, has revolutionized functional genomics [32]. The CRISPR/Cas9 system consists of a Cas9 endonuclease paired with a single guide RNA (sgRNA). The sgRNA sequence is used to target genomic DNA by complementary base pairing. When the sgRNA binds to the target DNA, the Cas9 protein induces double-stranded breaks (DSBs) in the genomic locus. The presence of DSBs initiates the cellular DNA repair pathways: non-homologous end joining (NHEJ) or homology directed repair (HDR) [33]. Repair through the NHEJ pathway can disrupt gene function through small base insertions or deletions, which may result in a frameshift mutation and loss of protein function (knockout). Co-expression of Cas9 and one or more gene-specific sgRNAs can thus result in gene knockout cells useful to assess gene function in various contexts. This gene-editing technology has allowed for the establishment of a few CRISPR/Cas9-edited mosquito cells lines, which includes an *Ae. aegypti*-derived Aag2 Dcr-2 knockout cell line [27], an Aag-2 Ago-2 knockout cell line [34], and an *Aedes albopictus*-derived C6/36 *Nix* knockout cell line [35]. However, these cell lines were established using a CRISPR/Cas9 plasmid originally designed for use in *D. melanogaster* (pAc-sgRNA-Cas9) [36]. There have been recent efforts to optimize CRISPR/Cas9-based gene editing in mosquito cell cultures. For instance, Anderson et al. 2020 [37] assessed the transcriptional activities of 33 Pol III promoters in *Ae. albopictus*-derived U4.4, *Ae. aegypti*-derived Aag2, and *Cx. quinquefasciatus*-derived Hsu mosquito cell lines. Rozen-Gagnon et al. 2021 [38] generated a plasmid-based CRISPR/Cas9 system for use in U4.4 and Aag2 cell populations. Viswanatha et al. 2021 [39] developed a genome-scale CRISPR screen using recombination-mediated cassette exchange (RMCE) to introduce the CRISPR/Cas9 reagents into *Anopheles coluzzii*-derived Sua5B cells. In Hsu cell lines, only one research group has constructed plasmids to deliver CRISPR/Cas9 components in cell cultures. Feng et al. 2021 [40] analyzed the activity of five insect promoters for Cas9 expression as well as six small nuclear RNA U6 promoters to drive sgRNA expression. However, the delivery of these CRISPR/Cas9 reagents involved co-transfection of three plasmids: one with a GFP reporter, one with the Cas9, and one with the sgRNA. The co-delivery of all CRISPR/Cas9 reagents in a single plasmid has been shown to increase gene-editing efficiency [38], yet a *Culex-*optimized plasmid that allows for robust expression of Cas9 and sgRNAs from a single construct remains lacking. 

In the present study, we generated a CRISPR/Cas9 construct containing *Culex*-optimized promoters to drive the expressions of Cas9 and sgRNAs in a single plasmid system based on previous work done by Feng et al. 2021 [40]. The newly designed *Culex*-optimized plasmid (pUb-Cas9-EGFP_cU6:6-sgRNA) showed superior in vitro gene editing when compared to the previously used *D. melanogaster* CRISPR/Cas9 construct. This plasmid demonstrated highly efficient editing of *dcr-2* and *piwi4* in Hsu cell cultures. The production of Hsu-derived clonal cells deficient in the RNAi proteins Dcr-2 and PIWI4 would be useful for investigating their roles in different contexts. This *Culex-*optimized CRISPR/Cas9 plasmid provides a significant tool for the identification of novel gene targets that may support, control, or mediate tolerance to virus replication in *Cx. quinquefasciatus* mosquitoes.

## 2. Materials and Methods

### 2.1. Cell Culture Maintenance

The *Cx. quinquefasciatus* ovary-derived (Hsu) cell line [41] was kindly provided by Dr. Aaron Brault (CDC for Vector-borne Diseases, Fort Collins) and grown at 27 °C and 5% CO_2_ in Dulbecco’s Modified Eagle Medium (DMEM; Corning #10-013-CV; Corning, NY, USA) supplemented with 10% FBS and antibiotics (100 units/mL penicillin, 100 μg/mL streptomycin, 5 μg/mL gentamicin).

### 2.2. Generation of Culex-Optimized CRISPR/Cas9 Plasmids

The *D. melanogaster* plasmid pAc-sgRNA-Cas9 [36] previously used to produce *Ae. aegypti*-derived (Aag2) Dcr-2 knockout cell line (AF319) [27] was obtained from Addgene (plasmid #49330). This plasmid relies on the *D. melanogaster* Actin-5c promoter to drive expression of the human codon-optimized *Streptococcus pyogenes* Cas9, and the *D. melanogaster* RNA Pol III U6:2 promoter to drive the expression of an sgRNA. The final *Drosophila*-optimized plasmid (pAc-Cas9-mCherry_dU6:2-sgRNA) was generated by replacing the puromycin resistance gene with a red fluorescent mCherry protein along with a self-cleaving T2A peptide to allow for efficient separation and coexpression of mCherry and Cas9 proteins.

The *Ae. aegypti* polyubiquitin promoter-driven Cas9 from Addgene plasmid #162161 [38], the enhanced green fluorescent protein (eGFP) sequence cloned from Addgene plasmid #32425 [42], and the *Cx. quinquefasciatus* RNA Pol III U6:6 promoter-driven sgRNA cloning site from Addgene plasmid #169323 [40] (kindly provided by Dr. Valentino Gantz) were assembled into a single plasmid using Gibson Assembly^®^ Cloning (New England Biolabs, #E5510S; Ipswich, MA, USA). The resulting plasmid (pUb-Cas9-EGFP_cU6:6-sgRNA) was generated to coexpress Cas9 endonuclease and eGFP via a self-cleaving T2A peptide along with a gene-specific sgRNA that can be easily inserted with BspQI restriction digest cloning using the same steps as the *Drosophila-*optimized plasmid. 

PCRs were carried out with Q5^®^ Hot Start High-Fidelity Master Mix (New England Biolabs, #M0494). Multiple overlapping amplicons were assembled using NEBuilder HiFi DNA Assembly Master Mix (New England Biolabs, #E2621; Ipswich, MA, USA) to generate the plasmids. Assembled plasmids were transformed into high-efficiency NEB Stable Competent *E. coli* (New England Biolabs, #C3040; Ipswich, MA, USA) for plasmid amplification. Plasmids were isolated using Zyppy Plasmid Miniprep Kit (Zymo Research, #D4019; Irvine, CA, USA) or ZymoPURE II Plasmid Maxiprep Kit (Zymo Research, #D4203; Irvine, CA, USA). The finalized constructs are shown in Figure 1. The sequences of all plasmids were verified using restriction digestion and Sanger sequencing.

### 2.3. sgRNA Design

CRISPR GuideXpress (https://www.flyrnai.org/tools/fly2mosquito/web/, accessed on 5 March 2021) was used to design the single guide RNAs (sgRNAs) targeting *dcr-2* or *piwi4*. The sgRNAs with the highest Housden efficiency score, off target score of 0, and coverage of <60% were selected. The sequences of the sgRNA oligonucleotides used for targeting *dcr-2* or *piwi-4* are indicated in Table 1.

### 2.4. Cloning of dcr-2 and piwi4 sgRNA Containing Plasmids

Gene-specific sgRNAs were inserted following the protocol previously described for pAc-sgRNA-Cas9 [38]. Briefly, the plasmids were first digested with BspQI (New England Biolabs, #R0712; Ipswich, MA, USA) to linearize in a 50 μL reaction (2 μg of plasmid, 5 μL 10× NEBuffer 3.1, 2μL BspQI, up to 50 μL with ddH_2_O). The linearized plasmids were visualized by agarose gel electrophoresis, and were subsequently isolated using a Zymoclean Gel DNA Recovery kit (Zymo Research, #D4002; Irvine, CA, USA). To prevent self-ligation following sgRNA insertion, the plasmid was dephosphorylated using Quick CIP (New England Biolands, #M0525; Ipswich, MA, USA). The reaction was stopped by heat-inactivation at 80 °C for 2 min. Complementary oligonucleotides of the sgRNA sequences were designed to lack the protospacer adjacent motif (PAM; NGG motif), and nucleotide sequences complementary to BspQI overhangs were added. The modified oligonucleotides are shown in Table 2. To anneal the two complementary sgRNA oligonucleotides, 5 μL of 100 μM forward and reverse oligonucleotides were combined with 10 μL of Quick Ligation Reaction Buffer (New England Biolabs, # B2200S; Ipswich, MA, USA). The oligonucleotides were denatured at 98 °C, and subsequently annealed using gradual cooling as shown in Table 3. The sgRNA inserts were then cloned into either *Drosophila-*optimized or *Culex*-optimized plasmids using Quick Ligase (New England Biolabs, #M2200S; Ipswich, MA, USA) and grown in high-efficiency NEB Stable Competent *E. coli* (New England Biolabs, #C3040; Ipswich, MA, USA) for plasmid amplification. Successful insertion of sgRNAs was confirmed using Sanger sequencing of all CRISPR/Cas9 plasmids.

### 2.5. Delivery of CRISPR/Cas9 Plasmid

Hsu cells were seeded in a 24-well plate prior to transfection. Hsu cells at 50–60% confluence were transfected with either *Drosophila*-optimized or *Culex*-optimized Cas9 plasmids expressing gene-specific sgRNAs using X-tremeGENE^™^ HP DNA transfection reagent (Roche, #6366236001; Pleasanton, CA, USA) as per manufacturer’s protocol for 24-well plate. Briefly, plasmid DNA concentration was normalized to 100 ng/μL. Then, 500 ng of plasmid was combined with 43 μL of Opti-MEM™ I Reduced Serum Medium (Thermo Fisher Scientific, #31985062; Waltham, MA, USA) prior to addition of 2 μL of X-tremeGENE^™^ HP DNA transfection reagent. Control-sgRNA treatment groups were transfected with Cas9 constructs containing nonspecific sgRNAs. Mock transfected cells were treated with Opti-MEM containing X-tremeGENE^™^ HP DNA transfection reagent alone. The lipoplex was incubated for 15 min at room temperature prior to addition to wells. Transfected cells were incubated at 27 °C and 5% CO_2_ in pre-warmed DMEM supplemented with 10% FBS and antibiotics (100 units/mL penicillin, 100 μg/mL streptomycin, 5 μg/mL gentamicin). All experiments were performed in duplicates. At three days post-transfection, the expression of Cas9 was confirmed using a Keyence fluorescence microscope and the presence of gene editing was confirmed using a T7 Endonuclease I (T7E1) assay.

### 2.6. Delivery of Synthetic sgRNA

Since using two distinct sgRNAs may increase editing efficiency [43,44,45], we included a synthetic sgRNA (sgRNA2; Table 1) targeting exon 3 of *dcr-2* that was procured from Synthego. We transfected the previously used *Culex*-optimized plasmid expressing *dcr-2* sgRNA1 into Hsu cells using X-tremeGENE^™^ HP DNA transfection reagent as described above. Immediately after addition of plasmid-containing lipoplex, cells were also transfected with the synthetic sgRNA2 using Lipofectamine™ RNAiMAX (Thermo Fisher Scientific, #13778150; Waltham, MA, USA) according to the manufacturer’s instructions. Briefly, 21.5 µL of Opti-MEM containing 17.5 µM of *dcr-2* sgRNA2 was combined with 21.5 µL of Opti-MEM with 2 µL of Lipofectamine™ RNAiMAX. The lipoplex was incubated for 5 min at room temperature prior to addition to wells. Transfected cells were incubated at 27 °C and 5% CO_2_. We used these two different transfection reagents, since we had previously tested and optimized reagents for plasmid and sgRNA transfections. The presence of gene editing was confirmed three days post-transfection using a T7E1 assay.

### 2.7. T7 Endonuclease I (T7E1) Assay

A T7 endonuclease assay was used to semi-quantitatively screen for gene editing efficiency as previously described [46,47]. T7 endonuclease detects any structural deformities in heteroduplexed DNA (e.g., mismatches/insertions/deletions) and cleaves at that site. When a gene from a mixed population of edited and non-edited cells is amplified by PCR, denatured, and annealed, heteroduplexes will form between PCR products from non-edited cells and the mix of various mutations/insertions/deletions that may have been created due to a Cas9-induced dsDNA break and NHEJ repair. Thus, the more edited the cell population is, the more product will be cleaved by T7 endonuclease due to heteroduplex formation. Briefly, genomic DNA (gDNA) was extracted at three days post-transfection using the Quick-DNA Miniprep kit (Zymo Research, #D3024; Irvine, CA, USA) as per manufacturer’s protocol. The genomic target loci were PCR-amplified using primers flanking the targeted gDNA (Table 4). Briefly, the PCR reactions were performed using Q5^®^ Hot Start High-Fidelity Master Mix (New England Biolabs, #M0494; Ipswich, MA, USA) and 100 ng of gDNA template in 50 μL of total volume. The PCR products were visualized by agarose gel electrophoresis, and the 900–1000 bps bands amplifying sgRNA-target regions were isolated using a Zymoclean Gel DNA Recovery kit (Zymo Research, #D4002; Irvine, CA, USA). For the T7EI assay, 200 ng of gel-purified PCR amplicons were mixed with 2 μL of 10× NEBuffer 2 (New England Biolabs, #B7002; Ipswich, MA, USA) and made up to 20 μL with ddH_2_O. The mixture was subsequently denatured for 5 min at 95 °C, and then re-annealed by gradual cooling to produce potential heteroduplexes of wild-type and mutated DNA strands where double-strand breaks have occurred. Then, 1 μL of T7 Endonuclease I enzyme (New England Biolabs, #M030; Ipswich, MA, USA) was added directly to the annealed PCR product and incubated at 37 °C for 15 min. The reaction was stopped by adding 1.5 μL of 0.25M EDTA, and the resulting products were resolved by agarose gel electrophoresis. All original gel images can be seen in Appendix A.

## 3. Results

### 3.1. Superior dcr-2 Editing in Hsu Cells Using a Culex-Optimized Plasmid

Following construction of the Cas9 constructs, the gene editing efficiency of the *Culex*-optimized plasmid (pUb-Cas9-EGFP_cU6:6-sgRNA) in Hsu cells was compared to the previously used *Drosophila*-optimized plasmid (pAc-Cas9-mCherry_dU6:2-sgRNA). The general workflow of the experimental setup is shown in Figure 2. Hsu cells were transfected with the *Culex* or *Drosophila* plasmids containing sgRNAs targeting either the *dcr-2* genomic sequence or a nonspecific control. At three days post-transfection, expression of Cas9 in plasmid-transfected cells was confirmed using fluorescence microscopy (Figure 3a). In all samples, high levels of fluorescence were detected and the percentage of transfected cells was comparable between plasmids (Figure 3a), especially for the transfection of plasmid containing *dcr-2* targeting sgRNA (33–36 % transfection efficiency). A T7EI assay was used to confirm gene-editing activity. A PCR amplicon of the expected size (857 bps) was observed in all samples (Figure 3b). No *dcr-2* editing was detected in the *Drosophila* plasmid-transfected cells, but two smaller bands indicative of T7EI digestion were detected in the *dcr-2* targeting *Culex* plasmid-transfected cells. These results show that transfection with the *Culex*-optimized plasmid allows for superior editing of the *dcr-2* gene in Hsu cells.

### 3.2. Transfection of Culex-Optimized CRISPR/Cas9 Construct and a Synthetic sgRNA Mediates Efficient Editing of dcr-2 Gene in Hsu Cells

The use of one single high-quality sgRNA can induce mutations resulting in gene knockout. However, using multiple sgRNAs can lead to higher efficiency of gene editing partly because it can lead to elimination of the intervening genomic region in at least a proportion of the cells [43,44,45]. To increase gene-editing efficiency of *dcr-2*, we transfected Hsu cells with a synthetic sgRNA (sgRNA2) targeting exon 3 of *dcr-2* following transfection with the *Culex-*optimized plasmid expressing sgRNA1 targeting exon 7 of the same gene. In parallel, Hsu cells were transfected with the *Culex-*optimized plasmid containing nonspecific sgRNA as a control. Following observation of eGFP-tagged Cas9 at three days post-transfection, a T7EI assay was performed on the gDNA from the transfected Hsu cells. PCR products of the expected sizes (sgRNA1 locus with 857 bps; sgRNA2 locus with 965 bps) were observed in all samples. In contrast to the control sgRNAs, which showed no detectable mutations with the T7EI cleavage assay, the PCR amplicons of sgRNA1-targeted and sgRNA2-targeted regions revealed two smaller T7E1-digested fragments (Figure 4). These results demonstrate that the transfection with multiple sgRNAs allows for Cas9-mediated gene editing at two different sites within the same gene.

Some nonspecific banding was observed in our control for the genomic region targeted by sgRNA2 (Figure 4, synthetic sgRNA), but this is common with T7 endonuclease assays due to naturally occurring SNPs and genetic variation in the culture. We confirmed that these nonspecific bands were not a result of our control sgRNA by repeating the assay on untreated Hsu gDNA, which showed the same nonspecific bands (Appendix A).

### 3.3. Editing of piwi4 Using the Culex-optimized CRISPR/Cas9 Construct in Hsu Cells

To validate the general applicability of our method, we applied our workflow to induce mutations in the *piwi4* gene in Hsu cells. First, two sgRNAs were designed to target the *piwi4* genomic region. Hsu cells were then transfected with *Culex*-optimized plasmid expressing sgRNAs targeting either exon 2 (sgRNA1), exon 4 (sgRNA2), or a nonspecific sgRNA as a control. After eGFP fluorescence was observed in Hsu cells at three days post-transfection, T7E1 assay was performed on the PCR-amplified target regions from the gDNA of transfected Hsu cells. All PCR amplicons showed expected sizes (sgRNA1 locus with 949 bps; sgRNA2 locus with 882 bps), as shown in Figure 5. No detectable mutations were observed in Hsu cells transfected with the *Culex-*optimized plasmid expressing control sgRNAs. However, the approximate fragment sizes of T7E1-digested amplicons were observed for sgRNA1-targeted as well as sgRNA2-targeted regions. These results suggest that the two sgRNAs induced gene editing at the targeted *piwi4* sites in Hsu cells.

## 4. Discussion

The present study demonstrated that the *Culex*-optimized construct induced mutations in *Cx. quinquefasciatus*-derived Hsu cell populations efficiently and reliably. We showed that driving the expression of Cas9 with the *Ae. aegypti* polyubiquitin promoter coupled with a *Cx. quinquefasciatus* U6:6 promoter to drive sgRNA expression exhibited superior in vitro gene editing when compared to the *D. melanogaster* Actin-5c and U6:2 promoters. These observations are consistent with the works of Anderson et al. 2020 [37] and Feng et al. 2021 [40], which reported that *D. melanogaster* promoter-driven Cas9 and sgRNA expression exhibited weak activity in Hsu cells. 

Our results demonstrated that the *Culex*-optimized plasmid performed efficient gene editing of multiple *dcr-2* and *piwi4* regions. Transfection of Hsu cells with *Culex*-optimized plasmid and an additional synthetic sgRNA, as well as co-transfection of two *Culex-*optimized plasmids containing different sgRNAs targeting the same gene both effectively induced mutations in the genomic regions of interest. These results suggest that this plasmid ensures robust coexpression of CRISPR/Cas9 components in a single plasmid, which is critical to achieving high editing efficiencies in Hsu cells. 

Future work will be aimed at generating Hsu-derived monoclonal knockout cell lines lacking functional Dcr-2 or PIWI4. We have already constructed a *Culex*-optimized plasmid expressing a red fluorescent protein (mCherry)-tagged Cas9 (instead of eGFP), allowing for the visualization of individual cells co-transfected with two separate sgRNAs through the detection of cells expressing both mCherry and eGFP. Distinct sgRNAs targeting the same gene will be cloned into these *Culex*-optimized plasmids, one sgRNA in the plasmid expressing eGFP-tagged Cas9 and the other sgRNA in the plasmid expressing mCherry-tagged Cas9. These plasmids will be co-transfected into Hsu cells and flow cytometry will be used to sort for cell singlets expressing both mCherry and eGFP fluorescent proteins. We anticipate that this method will further improve the probability of successful knockout of target genes in Hsu cells. 

Although the T7E1 assay is commonly performed to screen for the presence of mutations induced by CRISPR/Cas9, this method only provides a semi-quantitative measure of gene-editing efficiencies. In future studies, next generation and Sanger sequencing approaches will be used on monoclonal cell populations obtained via the methods described above. In addition, quantitative real-time PCR can provide valuable information on residual gene expression and presence of insertions or deletions as previously described by Yu et al. 2014 [48].

In summary, our results suggest that the designed construct efficiently expresses all CRISPR/Cas9 components under *Cx. quinquefasciatus*-optimized promoters in a single plasmid system. This construct expands the tools currently available for the generation of *Culex* mosquito-derived knockout cell lines and other genomic studies of this mosquito vector. Based on previous characterization of these promoters [37,38,40], we also anticipate this plasmid to be efficient for gene knockout in other mosquito cell lines, such as *Ae. aegypti*-derived Aag2 or *Cx. tarsalis*-derived CT cells. Our plasmids are available on Addgene (#190597 and #190598) for the scientific community and provide a valuable new resource to generate single or multigene knockout cells. Such knockout cell cultures will facilitate investigations into the interactions of mosquito cells with viral pathogens, which holds great potential to support the multifaceted effort to control arboviral diseases.

## Figures and Tables

**Figure 1 insects-13-00856-f001:**
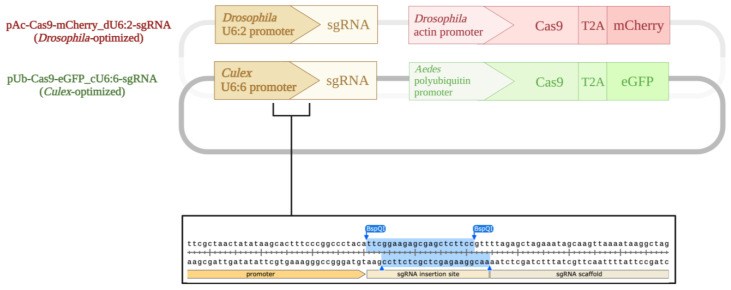
Plasmids used in this study. The pAc-sgRNA-Cas9 plasmid [36] was modified to encode for a Cas9 tagged with mCherry to generate the *Drosophila-*optimized CRISPR/Cas9 construct (pAc-Cas9-mCherry_dU6:2-sgRNA). The *Culex*-optimized plasmid (pUb-Cas9-EGFP_cU6:6-sgRNA) uses an *Ae. aegypti* polyubiquitin promoter to drive eGFP-tagged Cas9 expression and the *Cx. quinquefasciatus* RNA Pol III U6:6 promoter to drive sgRNA expression. Gene-specific sgRNA sequences can be inserted using simple BspQI restriction digest cloning. Created with BioRender.com.

**Figure 2 insects-13-00856-f002:**
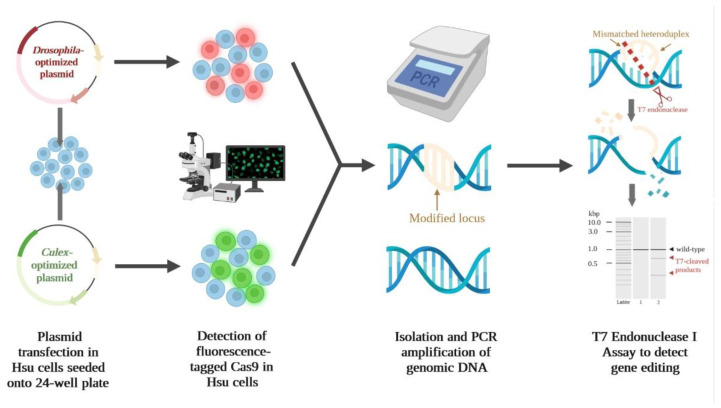
Detection of CRISPR/Cas9-induced mutations at the *dcr-2* locus in *Cx. quinquefasciatus*-derived Hsu cell line. Overview of the experimental workflow for validation of gene editing in plasmid-transfected mixed Hsu cell populations. Created with BioRender.com.

**Figure 3 insects-13-00856-f003:**
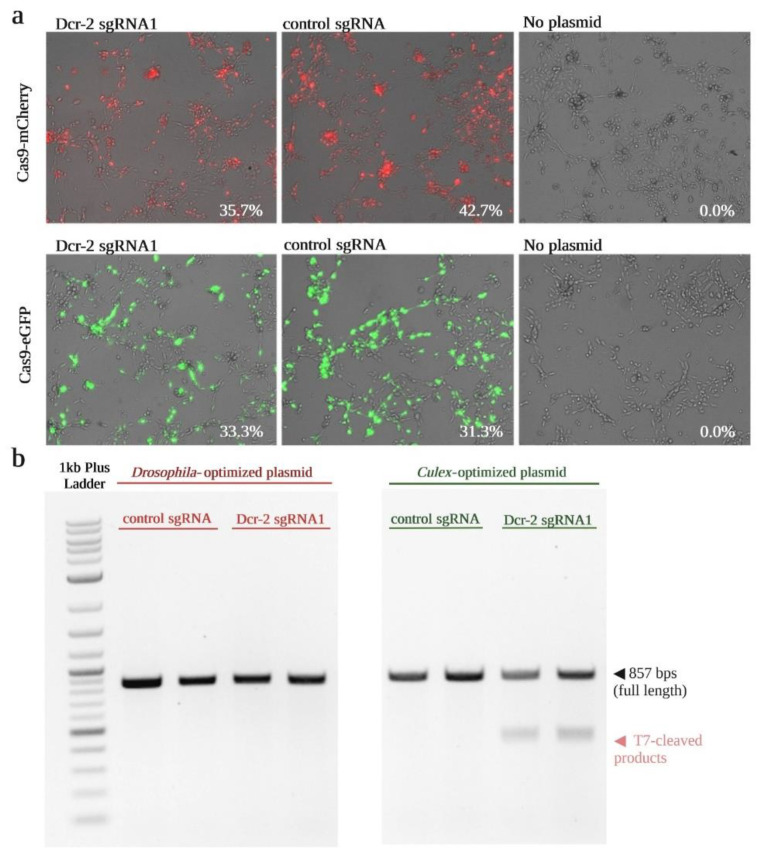
Detection of CRISPR/Cas9-induced mutations at the *dcr-2* locus in the *Cx. quinquefasciatus*-derived Hsu cell line. (**a**) Detection of fluorescence-tagged Cas9 at three days post-transfection. Hsu cells transfected with *Drosophila*-optimized plasmid show red fluorescent signal (mCherry) while the *Culex-optimized* plasmid shows green fluorescent signal (eGFP). The indicated percentage of fluorescent cells was calculated using the Keyence BZ-X analyzer software. (**b**) Agarose gel following T7EI assay was used to detect editing of *dcr-2* in Hsu cells three days post-transfection with the *Drosophila* plasmid (**left**) or the *Culex* plasmid (**right**), expressing either a control sgRNA or sgRNA1. The presence of smaller ~450 bps bands is indicative of gene editing. All treatments were performed in duplicate.

**Figure 4 insects-13-00856-f004:**
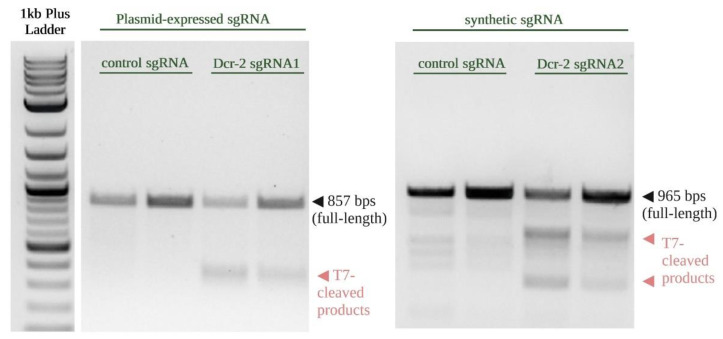
T7 Endonuclease I (T7E1) assay demonstrates genome editing in the *dcr-2*-targeted regions. Genomic DNA was isolated from Hsu cell populations three days post-transfection with the *Culex*-optimized plasmid expressing sgRNA1 and a synthetic sgRNA2. T7E1 assay detected editing at the genomic regions targeted by sgRNA1 (**left**) and sgRNA2 (**right**), as indicated by cleaved PCR products. All treatments were performed in duplicate.

**Figure 5 insects-13-00856-f005:**
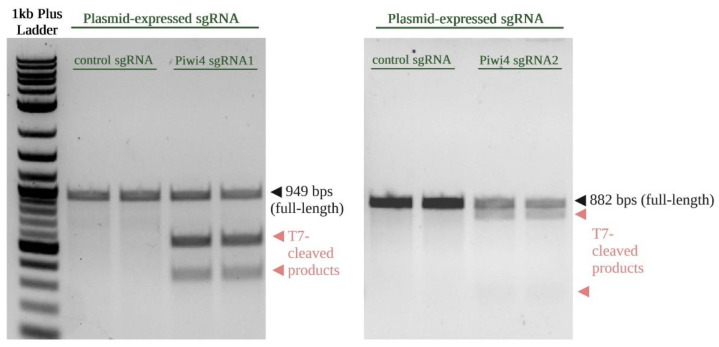
T7 endonuclease I (T7E1) assay for detection of CRISPR/Cas9-induced mutations at the *piwi4* locus in Hsu cells. T7E1 assay revealed mutations around the dsDNA break induced by Cas9 in the sgRNA1 (**left**) and sgRNA2 (**right**)-targeted *piwi4* regions, as indicated by presence of T7E1-digested fragments. All treatments were performed in duplicate.

**Table 1 insects-13-00856-t001:** sgRNA sequences targeting *dcr-2* and *piwi4*.

Gene (VectorBase ID)	sgRNA#	sgRNA Sequence-PAM (5′-3′)
*dcr-2* (CPIJ010534)	sgRNA1 sgRNA2	TACGTGCTGCGCATAGCGGCAGG CCCGACAGGCCAATCACCCGAGG
*piwi4* (CPIJ002459)	sgRNA1 sgRNA2	GAGCACAAGAAAATCTTCGGAGG CCTCGGCTGCATGATCCAGGCGG

**Table 2 insects-13-00856-t002:** Modified sgRNA oligonucleotides for cloning into CRISPR/Cas9 plasmids.

Gene (VectorBase ID)	sgRNA#	Oligonucleotides (with BspQI Compatible Overhangs *)
*dcr-2* (CPIJ010534)	sgRNA1	5′-TTCGTACGTGCTGCGCATAGCGGC-3′ 3′-AAGGCCGCTATGCGCAGCACGTAC-5′
	sgRNA2	5′-TTCGCCCGACAGGCCAATCACCCG-3′ 3′-AAGCGGGTGATTGGCCTGTCGGGC-5′
*piwi4* (CPIJ002459)	sgRNA1	5′-TTCGGAGCACAAGAAAATCTTCGG-3′ 3′-AACCCGAAGATTTTCTTGTGCTCC-5′
	sgRNA2	5′-TTCGCCTCGGCTGCATGATCCAGG-3′ 3′-AACCCTGGATCATGCAGCCGAGGC-5′

* compatible overhangs are shown as underlined sequences for each oligonucleotide sequence.

**Table 3 insects-13-00856-t003:** Oligonucleotide annealing program.

Temperature	Time/Cycling
98 °C	1 min
98–88 °C	5 s, decrease 0.1 °C/cycle × 99 cycles
88–78 °C	10 s, decrease 0.1 °C/cycle × 99 cycles
78–68 °C	10 s, decrease 0.1 °C/cycle × 99 cycles
68–58 °C	10 s, decrease 0.1 °C/cycle × 99 cycles
58–48 °C	10 s, decrease 0.1 °C/cycle × 99 cycles
48–38 °C	10 s, decrease 0.1 °C/cycle × 99 cycles
38–28 °C	10 s, decrease 0.1 °C/cycle × 99 cycles
28–18 °C	10 s, decrease 0.1 °C/cycle × 99 cycles
12 °C	Hold

**Table 4 insects-13-00856-t004:** PCR primer sequence for T7EI assay.

Name	Primer Sequence (5′-3′)
T7-Assay_Dcr2-sgRNA1_F	ATTGTGGTGGCCGTTTTGCT
T7-Assay_Dcr2-sgRNA1_R	ATGGCGGTACTGCTTCGCAT
T7-Assay_Dcr2-sgRNA2_F	CAAGCGCACCTTCTTCATCGTG
T7-Assay_Dcr2-sgRNA2_R	GCTTTGATCGACGAAAACAGCG
T7-Assay_Piwi4-sgRNA1_F	TTATAGCAGTGAGGGTCGTGAC
T7-Assay_Piwi4-sgRNA1_R	TTGCTTGTAGTACTCCACGAA
T7-Assay_Piwi4-sgRNA2_F	GTTCAGGCCGGTGAGCAG
T7-Assay_Piwi4-sgRNA2_R	GTGATCGAAGAAGCGCGTGTT

## Data Availability

The *Culex-*optimized plasmid pUb-Cas9-EGFP_cU6:6-sgRNA described here and its new mCherry counterpart pUb-Cas9-mCherry_cU6:6-sgRNA are available on Addgene as plasmids #190597 and #190598, respectively.

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
