# Peer review of "Optimized In Vitro CRISPR/Cas9 Gene Editing Tool in the West Nile Virus Mosquito Vector, Culex quinquefasciatus"

_insects, 2022, doi:10.3390/insects13090856_

Round 1
Reviewer 1 Report
The authors describe the development of a new tools for generating knockout Culex cell lines using CRISPR/CAS9. This is a fundamental step toward gene functional studies, particularly, as indicated in the text, those about immune response against arboviruses.
The manuscript is clear and the experimental design is linear and, in that, is fantastic and well performed.
I have only minor comments.
At the line 231 the authors describe the transfection of the plasmid and of an additional sgRNA. At the beginning it was not clear to me why they did it, till I read the results. Can the authors explain a little bit more why they did not transfect simultaneously the plasmid and the sgRNA in the same mix (using the same reagent) and can the give a little more details (how long they wait for the second transfection? Was it simultaneous but using two different transfection reagents?).
At line 270 the authors state that the percentage of transfected cells is comparable. Can they provide the data (as additional file)?
Gel Figures: they are nice and clear. Some of them are "composite", and it is fine to me. My only concern is about transparency, because there are mounting concerns about fairness of the data and about data replicability. I would suggest to provide the original gels in an additional file, so that if somebody need to check them, they are available and no discussion can arise.
For the rest, it is a long needed work that will with no doubts help improving our knowledge about the biology of Culex mosquitoes.
Reviewer 2 Report
This MS constructed a plasmid system containing Cas9 and sgRNAs, and transfected in the Cx. quinquefasciatus-derived Hsu mosquito cell lines for knocking-out dcr-2 and piwi4 genes. Generally speaking, this MS is well written and reasonable experimental design. However, authors only used T7EI assay to validate the editing of dcr-2 and piwi4 genes in Hsu Cells, and did no use sequencing method to confirm the mutation and other means to confirm the function loss of dcr-2 and piwi4 genes. So the conclusion “The present study demonstrated that the Culex-optimized construct induced mutations in Cx. quinquefasciatus-derived Hsu cell populations efficiently and reliably” is too strong. Also this MS more likely established a protocol for gene-editing in Hsu Cells instead of a research article.
(1) The cells in Figure 2b are not very clear, especially for the No sgRNA images, the cell shape is invisible. So it is hard to determine that Cas9 protein was expressed in the cell line. It is better to take these photos with high resolution again. Another alternative method is to add western blot results to confirm the expression of Cas9.
(2) Figure 2C, it appear that two gel images were put together instead one whole gel. It is better to show clear boundary.
(3) Feng et al, 2021 published two articles regrading gene knockout by CRISPR in Cx. Quinquefasciatus mosquito. What is the purpose to establish a new protocol in the cell line of Cx. Quinquefasciatus?
Reviewer 3 Report
1: The mechanism of T7 Endonuclease I (T7E1) assayperformed to screen mutations induced by CRISPR/Cas9in this work has not been explained or given any reference, which should be added to improve understanding of the results.
2: In figure 3, it is not explained why the control sgRNA corresponds to multiple bands. If it is accidental, the experiment should be repeated.
3: The T7E1 assay only provided a semi-quantitative result of the gene-editing efficiencies, sequencing of the genomic DNA target sites is needed to show the intuitive mutant sequence information.
4: The importance of Dicer2 and Piwi4 in the antiviral process of mosquitos has always emphasized (in abstract, introduction and discussion), and the aim of this work is to investigate mosquito host genes involved in pathogen interactions (in abstract (line 30-31), and introduction), thus, why not perform the virus infection test using the mosquito cell mutation? With the phenotype, it can fully reflect the value and significance of the optimizedCRISPR/Cas9 gene editing system in Culex quinquefasciatus.
Reviewer 4 Report
The authors optimized CRISPR/Cas9 gene editing tool Culex quinquefasciatus cells,this will supported the research of identification of mosquito host genes involved in antiviral response and may contributes to the development of gene-based vector control strategies. The language, data and conclusion are well presented. I think this manuscript will has a broad readers who are interested in mosquito gene editing or gene functional research.
Round 2
Reviewer 2 Report
All comments were addressed properly. I agree to accept.